# CONTEXT-AWARE ONLINE RECOMMENDATION WITH BAYESIAN INCENTIVE COMPATIBILITY

## ABSTRACT

Recommender systems play a crucial role in internet economies by connecting users with relevant products or services. However, designing effective recommender systems faces two key challenges: (1) the exploration-exploitation trade-off in balancing new product exploration against exploiting known preferences, and (2) context-aware Bayesian incentive compatibility in accounting for users' heterogeneous preferences and self-interested behaviors. This paper formalizes these challenges into a Context-aware Bayesian Incentive-Compatible Recommendation Problem (CBICRP). To address the CBICRP, we propose a two-stage algorithm (RCB) that integrates incentivized exploration with an efficient offline learning component for exploitation. In the first stage, our algorithm explores available products while maintaining context-aware Bayesian incentive compatibility to determine sufficient sample sizes. The second stage employs inverse proportional gap sampling integrated with arbitrary efficient machine learning method to ensure sublinear regret. Theoretically, we prove that RCB achieves $O(\sqrt{KdT})$ regret and satisfies Bayesian incentive compatibility (BIC). Empirically, we validate RCB's strong incentive gain, sublinear regret, and robustness through simulations and a real-world application on personalized warfarin dosing. Our work provides a principled approach for incentive-aware recommendation in online preference learning settings.

## 1 INTRODUCTION

In the current era of the internet economy, recommender systems have been widely adopted across various domains such as advertising, consumer goods, music, videos, news, job markets, and travel routes (Koren et al., 2009; Li et al., 2010; Covington et al., 2016; Wang et al., 2017; Zheng et al., 2018; McInerney et al., 2018; Naumov et al., 2019; Lewis et al., 2020; Bao et al., 2023). Modern recommendation markets typically involve three key stakeholders: products, users, and the platform (which acts as a *principal*). The platform collects and analyzes user data to enhance future distribution services and to respond effectively and promptly to user feedback. In these context-aware markets, the platform serves as the planner and fulfills a dual role: recommending the best available product (i.e., exploitation) and experimenting with lesser-known products to gather more information (i.e., exploration). This exploration is crucial because users often have heterogeneous preferences, and many products may initially seem unappealing. However, exploration feedback can be valuable it provides critical insights into the products and helps determine whether they might be worthwhile for future users with similar interests. Unlike in service-oriented scenarios, these are marketplaces where choices are ultimately made by users rather than imposed by the platform.

The key challenge arises from the fact that heterogeneous users exhibit various interests to exploration and are usually lack *incentives* to adhere to the platform's recommendations due to varying interests. A myopic user is likely to choose products based solely on immediate benefits, demonstrating a bias toward exploitation over exploration. How can the platform effectively keep a balance between exploration and exploitation while taking individualized incentive compatibility into account? In other words, recommender systems commonly face two significant obstacles: (1) *exploration-exploitation tradeoff*: How can the platform design recommender systems that maximize rewards but also consider that failing to sufficiently explore all available products initially may lead to suboptimal decisions? (2) *context-aware incentive compatibility*: How can we strategically address the tendency of heterogeneous users to behave myopically?

In this paper, we first formalize those challenges into a *Context-aware Bayesian Incentive-Compatible Recommendation Problem* (CBICRP). This protocol assumes that the platform can communicate directly with users, for example, by sending individualized product recommendations, and then observing the user's actions and the outcomes. The key difference between this protocol and standard bandit algorithm is that user's actions incorporate not only their personalized interests and a common public prior over all products but also the individualized message sent by the platform. That is, users will continuously evaluate the difference between products after receiving the message/recommendation sent by the platform which is formalized in a Bayesian way.

The basic multi-armed bandit (MAB) model of incentivized exploration has been examined in (Kremer et al., 2014; Che & Hörner, 2018; Mansour et al., 2020; Sellke & Slivkins, 2023), which model the recommendation policy within the framework of MAB problems and incorporating incentive compatibility constraints by agents' Bayesian priors, but these models assume independent prior preference over products but in reality, these products share correlated prior beliefs. Subsequently, Hu et al. (2022); Sellke (2023) propose BIC recommendation policies for customers with dependent priors with Thompson sampling algorithm. However, Hu et al. (2022) considered the combinatorial semi-bandits which didn't consider the users' contextual information and corresponding personalized preferences over products. Similarly, Sellke (2023); Kalvit et al. (2024) considered the fixed design setting where feature $x_i$ are product-owned and fixed rather than our setting that feature $x_{i,t}$ is user-possessed and online sampled which introduces more technical difficulty since fixed design of $x$ can be transformed into the MAB setting and randomized design of $x$ can not. In addition, their settings only need to learn one parameter and our setting needs to learn $K$ arms' parameters (Lattimore & Szepesvári, 2020; Bastani & Bayati, 2020). Besides, our framework can easily incorporate any efficient offline marching learning methods, which greatly strengthens its applicability.

Recommendation context bandit algorithm (RCB) is composed of a two-stage design's algorithm. In the first stage, the platform explores all available products, taking into account context-aware incentive compatibility, and determines the minimal amount of information (sample size) that needs be collected for the subsequent stage. The second stage employs an *inverse proportional gap sampling bandit* integrated with any efficient plug-in offline machine learning method. This approach aims to simultaneously ensure sublinear regret and maintain context-aware BIC.

Our main contributions can be delineated into three parts:

1. We formalize the context-aware online recommendation problem under BIC constraints in §3. This formulation accommodates context-aware user preferences and incorporates BIC constraints.

2. We introduce a two-stage context-aware BIC bandit algorithm (RCB) for addressing CBICRP (see Algorithms 1 and 2). This algorithm adapts to *any efficient offline machine learning algorithm* as a component of the exploitation stage. RCB is also a decision length $T$-free algorithm, as long as $T$ is greater than a constant. Moreover, we demonstrate that RCB achieves an $\mathcal{O}(\sqrt{KdT})$ regret bound (Theorem 2), where $K$ is the number of products and $d$ is the feature dimension. It also maintains the BIC constraints (Theorem 1).

3. Lastly, we validate the effectiveness of RCB through its performance in terms of incentive gain and sublinear regret, and its robustness across various environmental and hyperparameter settings in §6.1. Additionally, we apply our algorithm to real-world data (personalized warfarin dose allocation) and compare it with other methods to demonstrate its efficacy in §6.2.

In §2, we provide related works. In §3, we introduce the heterogeneous recommendation protocol featuring BIC and the associated challenges. §4 details the design of our algorithm. In §5, we demonstrate that RCB upholds the BIC constraint and suffers sublinear regret. §6 showcases the effectiveness and robustness of RCB through simulations and real-data studies.

**Notations.** We denote $[N] = [1, 2, ..., N]$ where $N$ is a positive integer. Define $x \in \mathbb{R}^d$ be a $d$-dimensional random vector. The capital $X \in \mathbb{R}^{d \times d}$ represents a $d \times d$ real-valued matrix. Let $I_d$ represent a $d \times d$ diagonal identity matrix. We use $\mathcal{O}(\cdot)$ to denote the asymptotic complexity. We denote $T$ as the time horizon.

## 2 RELATED WORKS

**Incentivized Exploration**. There is a growing literature about a three-way interplay of exploration, exploitation, and incentives, comprising a variety of scenarios. The study of mechanisms to incentivized exploration has been initiated by (Kremer et al., 2014). They mainly focus on deriving the Bayesian-optimal policy for the case of only two actions and deterministic rewards, where Che & Horner (2015) also propose a model to derive a BIC policy to this setting. Frazier et al. (2014) considers a different setting with monetary transfers between the platform and agents. Later, exploration-exploitation problems with multiple self-interested agents have also been studied: multiple agents engaging in exploration without a planner to coordinate them e.g., (Keller et al., 2005), context-aware pricing with model uncertainty e.g., (Besbes & Zeevi, 2009; Badanidiyuru et al., 2018), dynamic auctions e.g., (Ostrovsky & Schwarz, 2023; Han & Dai, 2023), pay-per-click ad auctions with unknown click probabilities e.g., (Babaioff et al., 2015), as well as human computation e.g., (Ho et al., 2014).

**Bandit Algorithms**. There are various strategies and algorithms to solve the sequential decision making problem (Bubeck et al., 2012; Slivkins et al., 2019; Maillard, 2019; Lattimore & Szepesvári, 2020), such as the $\epsilon$-greedy (Auer et al., 2002; Chen et al., 2021; Han et al., 2022; Shi et al., 2022), explore-then-commit (Robbins, 1952; Abbasi-Yadkori et al., 2009; Li et al., 2022), upper confidence bound (UCB) (Lai & Robbins, 1985; Auer, 2002; Li et al., 2021; Wang et al., 2023), Thompson sampling (Thompson, 1933; Russo & Van Roy, 2014; Li et al., 2023), boostrap sampling (Kveton et al., 2019; Wang et al., 2020; Wu et al., 2022; Ramprasad et al., 2023), information directed sampling (Russo & Van Roy, 2014; Hao & Lattimore, 2022), inversely proportional to the gap sampling (Abe & Long, 1999; Foster & Rakhlin, 2020; Simchi-Levi & Xu, 2022), and betting (Waudby-Smith et al., 2022; Li et al., 2024). Additional related works can be found in Appendix §A.

## 3 RECOMMENDATION PROTOCOL

We first illustrate the basic *Context-aware Bayesian Incentive-Compatible Recommendation Problem* (CBICRP). Assume a sequence of $T$ streaming users arrive sequentially to the platform and each user $p_t$ with covariates (features) $x_t$ such as age, race, and location where these observed covariates $\{x_t\}_{t\geq 1}$ are drawn independently from distribution $\mathcal{D}_X$ over a deterministic set $\mathcal{X} \subset \mathbb{R}^d$. The platform has a set of products $\mathcal{A}$, e.g., ads/music/video/medicine, where $|\mathcal{A}| = K$. Each product (also called as arms in bandit literature) is represented as the *unknown* vector $\beta_i \in \mathbb{R}^d$. At time $t$, user $p_t$ arrives at the platform and the platform need to recommend arms to the user which follows the following protocol:

1. The platform recommends the user with a best arm $I_t$ based on user's covariates $x_t$.

2. User myopically chooses an action $a_t \in \mathcal{A}$ and receives a stochastic reward $y_t(a_t) \subset \mathcal{Y}$ where $\mathcal{Y} \in [0, 1]$, and leaves.

3. We assume the user provides reward $y_t(a_t)$ following the linear model $y_t(a_t) = \mu(x_t, a_t) + \eta_{t,a_t}$, where $\mu(x_t, a_t) = x_t^{\mathsf{T}} \beta_{a_t}$.[1]

and $\{\eta_{t,a_t}\}_{t\geq 1}$ are $\sigma$-subgaussian random variables if $\mathbb{E}[e^{t\eta}] \leq e^{\sigma^2 t^2/2}$ for every $t \in \mathbb{R}$, and independent of the covariates $\{x_t\}_{t\geq 1}$. Besides, for notation simplicity, let $y_t$ denote the vector potential reward in $[0, 1]^K$, $\mu(x_t)$ as the vector true personalized reward in $[0, 1]^K$, and $\eta_t$ as the vector noise in $\mathbb{R}^d$. Without loss of generality, we assume $\mathcal{X}$ and $\beta$ are bounded which means that it exists positive constants $x_{\max}$ and $b$ such that $\|x\|_2 \leq L, \forall x_t \in \mathcal{X}$ and $\|\beta_i\|_2 \leq b$ for all $i \in [K]$, which is a common assumption in literature (Abbasi-Yadkori et al., 2011; Bastani & Bayati, 2020; Li et al., 2021) and usually assume $L = b = 1$. It's important to note that the reward function contains two stochastic sources: the covariate vector $x_t$ and the noise $\eta_t$, which is general harder than the fixed design $\{x_t\}_{t\geq 1}$ in bandit (Lattimore & Szepesvári, 2020). Besides, we define the data domain $\mathcal{Z} = \mathcal{X} \times \mathcal{Y}$ and denote $\mathcal{D}_{\mathcal{Z}}$ as the probability distribution over set $\mathcal{Z}$.

The key difference between the above recommendation protocol with previous literature in sequential decision making (Sutton & Barto, 2018; Lattimore & Szepesvári, 2020) is that the user $p_t$ *may not follow the (best) recommendation arm* $I_t$, that is, $I_t \neq a_t$. Users can switch to other recommended options rather than simply click or not click the best recommended product provided by

---

[1] The discussion of the nonlinear is available in Appendix §F.

the algorithm. However, in CBICRP, the platform performs as a principal to recommend $I_t$ and the decision $a_t$ is made by the user based on prior knowledge over arms, and the user have the option to other products recommended by the platform. We assume the platform and all users share a prior belief over arms $\mathcal{P}_0 = \mathcal{P}_{1,0} \times ... \times \mathcal{P}_{K,0}$ where product prior parameter $\beta_i \sim \mathcal{P}_{i,0}$ with the mean $\beta_{i,0} = \mathbb{E}[\beta_i]$ and covariance matrix $\mathrm{var}(\beta_i) = \Sigma_{i,0}$. Additionally, given covariate $x_t$, denote $\mu_0(x_t, i) = \mathbb{E}[\mu(x_t, i)]$ as the prior mean reward for arm $i$. It's important to note that this setting is different from the bandit setup whose arm parameter $\beta_i$ is unknown and fixed.

Ideally, we hope users follow the (best) recommended arm $I_t$ even it is not the greedy option for them given that the goal of each user is to maximize her expected reward conditional on her priors over products. Here we define the event that best recommendations have been followed in the past before time $t$ with prior knowledge $\mathcal{P}_0$ as $\Gamma_{t-1} = \{I_s = a_s : s \in [t-1]\} \cup \mathcal{P}_0$, which works as a public information. Then we can formally define the $\epsilon$-*Context-aware Bayesian-Incentive Compatible* (CBIC) for users as follows.

**Definition 1** ($\epsilon$-CBIC). Given an *incentive budget* $\epsilon \geq 0$, a recommendation algorithm is $\epsilon$-Context-aware Bayesian Incentive-Compatible ($\epsilon$-CBIC) if

$$\mathbb{E}[\mu(x_t, i) - \mu(x_t, j)|I_t = i, \Gamma_{t-1}] \geq -\epsilon, \quad \forall t \in [T], i \in [K]. \tag{1}$$

If $\epsilon = 0$, we call it Context-aware Bayesian Incentive-Compatible (CBIC). For brevity, we use the term CBIC to denote both CBIC and $\epsilon$-CBIC throughout the following paper, unless emphasized.

This definition implies that after receiving additional information, such as the recommended arm $I_t$ and the historical information $\Gamma_{t-1}$, the user always follow the recommended arm or at most with expected reward (informally speaking, utility) loss less than $\epsilon$. Specifically, the user selects the arm $i$ that maximizes the posterior mean reward, which is either the best recommended arm $I_t$ or another arm whose posterior mean reward is within an $\epsilon$ budget of the maximum. From the perspective of the principal, it needs to *contextually* determine which arm to be recommended based on the current covariate $x_t$ and all historical feedback $\mathfrak{S}_{1:t-1}$ at time $t$, where $\mathfrak{S}_{1:t} = \{(x_t, y_t, a_t)\}_{1:t}$ denotes the sigma-algebra generated by the history up to round $t$. The objective for the platform is to design a sequential decision-making policy $\pi = \{\pi_t(\cdot)\}_{t \geq 1}$ that maximizes the expected reward for each user while adhering to the CBIC constraint, where $\pi_t(x_t|\mathfrak{S}_{1:t-1}) : \mathcal{X} \to \mathcal{A}$ denote the arm chosen at time $t$. Finally, let's define the regret with respect to CBIC constraint when following the policy $\pi$. The regret $\lambda_{[T]}(\pi)$ is defined as follows:

$$\mathrm{Reg}_{[T]}(\pi) = \sum_{t=1}^{T} \mathbb{E}\big[\mu(x_t, \pi_t^*(x_t)) - \mu(x_t, \pi_t(x_t))\big] \tag{2}$$

where $\pi_t^*(x_t)$ is the posterior optimal arm given all information up to $t - 1$, covariate $x_t$, and prior knowledge $\mathcal{P}_0$. The $\mathrm{Reg}_{[T]}(\pi)$ is taken over the randomness in the realized rewards and the randomness inherent in the algorithm. Finally, we summarize the key challenge in the CBICRP:

---
**Key challenge:**

In CBICRP, users exhibit context-aware prior preferences over arms, requiring that recommended products be more valuable than those selected myopically even still within an $\epsilon$ margin of the maximum reward. Concurrently, the platform aims to maximize long-term expected rewards. Therefore, the principal challenge lies in designing an algorithm that can **simultaneously** balance the users' incentive, the platform's requirement for maximizing expected rewards, and the exploration.

---

## 4 ALGORITHMS

In this section, we introduce the Recommendation Contextual Bandit (RCB) algorithm, which is structure into two stages, the *cold start* stage and the *exploitation* stage. The objective during the cold start stage is to develop an algorithm that not only maintains CBIC for users to gain trust but also fulfills the minimal sample size requirement necessary for the subsequent algorithm requirement for the platform with minimal budget and time cost. In the second stage, the design of RCB focuses on constructing a sampling bandit algorithm that incorporate any efficient offline machine learning methods for the long term goal of the balance of freshness and exploitation. This goal is fulfilled by balancing the $\epsilon$-budget allocation strategically and a carefully designed of sequential spread parameter $\{\gamma_m\}_m$ over algorithm's batches $m$.

## 4.1 COLD START STAGE

During the cold start stage, it needs to determine two important quantities, *minimum sample size* $\mathsf{N}$ for each arm and *exploration probability* $L$. In addition, denote $N_i(t)$ as the current number of pulls of arm $i$ at time $t$, and $B_t = \{i \mid N_i(t) = \mathsf{N}, \forall i \in [K]\}$ as the set of arms that have been pulled $\mathsf{N}$ times. Additionally, $S_i$ represents the set collecting historical rewards and covariates for arm $i$, and $S = \{S_k\}_{k \in [K]}$ encompasses the historical information for all arms.

The cold start stage's process comprises two steps: (1) identify the most popular arm based on the context-aware preference priors, and (2) recommend the remaining arms in a manner that economically allocates the incentive budget.

**(1) The Most Popular Arm's Sample Collection (MPASC).** If no arm has collected $\mathsf{N}$ samples, meaning $B_t$ is empty, the platform recommends arm $i$ to agent $p_t$, where arm $i$ has the highest prior mean reward with respect to agent $p_t$. Subsequently, agent $p_t$ provides feedback $y_{t,i}$ according to reward model. Afterwards, the platform updates the number of pulls $N_i(t)$ and the data $S_i$ respectively: $N_i(t) = N_i(t-1) + 1, S_i = S_i \cup (x_t, y_{t,i})$. Once an arm has been pulled $\mathsf{N}$ times, it is removed from further consideration and added to $B_t$. The principle initially verifies whether any arm has accumulated $\mathsf{N}$ samples. This step determines which arm is prior optimal, indicating the most popular among heterogeneous users.

**(2) Rest Arm Sample's Collection (RASC).** The platform initially samples a Bernoulli random variable $q_t \sim \text{Ber}(1/L)$ to determine the recommendation strategy for the current user. With a probability of $1/L$, the platform recommends exploring promoted (sample-poor) products, while with an exploitation probability of $1 - 1/L$, it suggests exploiting organic (sample-efficient) products. The optimal value of $L$ is determined based on prior information and the incentive budget $\epsilon$, as specified in Theorem 1 in §5.

**a) Promoted Recommendation.** If $q_t = 1$, the platform recommends agent $p_t$ to explore with a promoted arm which is the highest prior mean reward arm within the set of $[K]/B_t$, representing that arms have not been pulled $\mathsf{N}$ times,

$$\widetilde{a}_t = \underset{i \in [K]/B_t}{\operatorname{argmax}} \mathbb{E}[\mu(x_t, i)]. \tag{3}$$

Then agent $p_t$ receives reward $y_{t,\widetilde{a}_t}$ and the platform updates the number of pulls and samples of pair of the covariate and reward respectively: $N_{\widetilde{a}_t}(t) \leftarrow N_{\widetilde{a}_t}(t-1) + 1, S_{\widetilde{a}_t} \leftarrow S_{\widetilde{a}_t} \cup (x_t, y_{t,\widetilde{a}_t})$. When arm $\widetilde{a}_t$ has been pulled $\mathsf{N}$ times, arm $\widetilde{a}_t$ is added to set $B_t$.

**b) Organic Recommendation.** If $q_t = 0$, the platform recommends the agent $p_t$ to exploit with the organic arm $a_t^*$, which is the highest expected mean reward arm conditional on $S_{B_t}$.

$$a_t^* = \underset{i \in [K]}{\operatorname{argmax}} \mathbb{E}[\mu(x_t, i)|S_{B_t}]. \tag{4}$$

That is, arms in $B_t$'s expected rewards are evaluated through posterior mean rewards and arms not in $B_t$'s expected rewards are evaluated through prior mean rewards. Then the agent $p_t$ receives reward $y_{t,a_t^*}$, but in this case, the principal will not update $N_{a_t^*}(t)$ and $S_{a_t^*}$.

## 4.2 EXPLOITATION STAGE

Given the data $S$ (defined in §4.1) collected during the cold start stage, where each arm accumulates $\mathsf{N}$ samples, the platform's objective in the exploitation stage is to recommend arms with higher posterior means while satisfying the CBIC constraint. Thus, the key challenge of the bandit algorithm's design lies in balancing exploitation efficiency with the allocation of the incentive budget $\epsilon$. The general principle of the bandit algorithm involves first strategically dividing the decision points into a series of epochs of increasing length. At the beginning of each epoch, samples collected in the previous epoch are used to update the *spread parameter* $\gamma_m$ to control the balance of exploration and exploitation tradeoff at epoch $m$, thereby informing the decisions for the current epoch. Here we first denote the $m$th epoch's rounds as $\mathcal{T}_m = \{t \in [2^{m-1}, 2^m), m \geq m_0\}$ and $m(t)$ representing the epoch where the current $t$ belongs to. The cold start stage's epoch is demoted as $m_0 = \lceil 2 + \log_2 \mathsf{N} \rceil$ and the final stage is denoted as $m_1$. The principal collected data at the $m$th epoch denoted as $W_{\mathcal{T}_m} = \{x_t, a_t, y_t(a_t)\}_{t \in \mathcal{T}_m}$.

At epoch $m \in [m_0, m_1]$, the platform then obtains the posterior mean estimator $\widehat{\beta}_i = \mathbb{E}_{\beta_i \sim p(\beta_i | W_{\mathcal{T}_{m-1}})}[\beta_i]$, where $p(\beta_i | W_{\mathcal{T}_{m-1}})$ represents the posterior distribution based on data from

---

**Algorithm 1:** `Cold Start Stage`

**Input** : $K, \mathsf{N}, L, B, S, \{N_i(t)\}_{i \in [K]}, t = 1$.

1 STEP 1 - THE MOST POPULAR ARM SAMPLE COLLECTION (MPASC)

2 **while** *there is no arm been pulled* $\mathsf{N}$ *times* **do**

3     Agent $p_t$ is recommended with arm $i = \operatorname{argmax}_{j \in [K]} \mathbb{E}[\mu(x_t, j)]$ and receives reward $y_{t,i}$.

4     The platform updates pulls and rewards: $N_i(t) \leftarrow N_i(t-1) + 1, S_i \leftarrow S_i \cup (x_t, y_{t,i})$.

5     If $N_i(t) = \mathsf{N}$, add $i$ to $B_t$. $t \leftarrow t + 1$. STEP 1 stopped.

6     Update $t \leftarrow t + 1$.

7 STEP 2 - REST ARM SAMPLE COLLECTION (RASC)

8 **while** *there exists an arm $i$ such that the number of pulled $N_i(t)$ has not reached* $\mathsf{N}$ **do**

9     Samples $q_t \sim \text{Ber}(1/L)$.

10     **if** $q_t = 1$ **then**

11        $p_t$ is recommended to explore with the arm $\widetilde{a}_t$ based on Eq.3 and receives $y_{t,\widetilde{a}_t}$.

12        Updates $N_{\widetilde{a}_t}(t) \leftarrow N_{\widetilde{a}_t}(t-1) + 1$ and dataset $S_{\widetilde{a}} \leftarrow S_{\widetilde{a}} \cup (x_t, y_{t,\widetilde{a}_t})$.

13        If $N_{\widetilde{a}_t}(t) = \mathsf{N}$, add $\widetilde{a}_t$ to $B_t$.

14     **else**

15        $p_t$ is recommended to exploit with the arm $a_t^*$ based on Eq.4 and receives $y_{t,a_t^*}$.

16     Update $t \leftarrow t + 1$.

---

---

**Algorithm 2:** `Exploitation Stage`

**Input** : $S$, epochs $m_0, m_1$, function class $\mathcal{F}$, learning algorithm $\mathrm{Off}_{\mathcal{F}}$, confidence level $\delta$.

1 **for** *epoch* $m \in [m_0, m_1]$ **do**

2     Set $\gamma_m = 4\sqrt{K/\mathcal{E}_{\mathcal{F},\delta}(|\mathcal{T}_{m-1}|)}$.

3     Feed $m - 1$ epoch's data $W_{\mathcal{T}_{m-1}}$ into the $\mathrm{OffPos}$ and get $\{\widehat{\beta}_{m,i}\}_{i \in [K]}$.

4     **for** $t \in \mathcal{T}_m$ **do**

5        Agent $p_t$ arrives with covariate $x_t$. Compute estimate $\widehat{\mu}_{m(t)}(x_t, i) = x_t^{\top} \widehat{\beta}_{m,i}, \forall i \in [K]$.

6        Obtain the optimal arm $b_t = \operatorname{argmax}_{i \in [K]} \widehat{\mu}_{m(t)}(x_t, i)$.

7        Sample $a_t \sim p_m(i)$ according to Eq.5 and observe reward $y_t(a_t)$.

---

$W_{\mathcal{T}_{m-1}}$). Subsequently, the platform computes the *predictive estimate reward* $\widehat{\mu}_t(x_t, i) = x_t^{\top} \widehat{\beta}_i$ for all arms. We denote $b_t = \operatorname{argmax}_{i \in [K]} \widehat{\mu}_t(x_t, i)$ as the *best predictive arm*. The platform then randomly selects arm $a_t$ according to the distribution $p_t(i)$, for $t \in \mathcal{T}_m$:

$$p_m(i) = \begin{cases} 1 - \sum_{i \neq b_t} p_t(i), & \text{if } i = b_t. \\ 1/[K + \gamma_m(\widehat{\mu}_t(x_t, b_t) - \widehat{\mu}_t(x_t, i))], & \text{if } i \neq b_t. \end{cases} \tag{5}$$

where the spread parameter $\gamma_m = 4\sqrt{K/\mathcal{E}_{\mathcal{F},\delta}(|\mathcal{T}_{m-1}|)}$ regulates the balance between exploration and exploitation, and $\mathcal{E}_{\mathcal{F},\delta}(|\mathcal{T}_{m-1}|)$ denotes the mean squared prediction error (MSPE) at epoch $m - 1$. A smaller $\gamma_m$ results in a more dispersed $p_t$, enhancing exploration. Conversely, a larger $\gamma_m$ leads to a more concentrated $p_t$, focusing recommendations on the best predictive arm $b_t$. As the epoch progresses, $\gamma_m$ increases and is inversely proportional to the square root of the MSPE. The MSPE is typically derived via cross-validation using an efficient offline statistical learning method. Below, we present the formal definition of $\mathcal{E}_{\mathcal{F},\delta}(n)$ with $n$ i.i.d. training samples.

**Definition 2.** Let $p$ be an arbitrary action selection kernel. Given a sample size of $n$ data of the format $(x_i, a_i, y_{i,a_i})$, which are i.i.d. according to $(x_i, y_i) \sim \mathcal{D}, a_i \sim p(\cdot|x_i)$, the offline learning algorithm $\mathrm{Off}_{\mathcal{F}}$ based on the data and a general function class $\mathcal{F}$ returns a predictor $\widehat{\mu}_t(x, a) : \mathcal{X} \times \mathcal{A} \to \mathbb{R}$. For any $\delta > 0$, with probability at least $1 - \delta$, we have $\mathbb{E}_{x \sim \mathcal{P}_X, a \sim p(\cdot|x)}[\widehat{\mu}_t(x, a) - \mu(x_t, a_t)]^2 \leq \mathcal{E}_{\mathcal{F},\delta}(n)$.

**Computational Cost**: The cold start stage's computational cost is $\mathcal{O}(KLN)$ in expectation and the exploitation stage's computational cost are mainly based on the offline sample efficient machine learning method. Usually it needs $\mathcal{O}(K/\epsilon'^2)$ samples in expectation for non-parametric methods and $\mathcal{O}(Kd/\epsilon')$ samples in expectation for parametric methods to get the desired offline error $\epsilon'$.

## 5 THEORY

In this section, we first provide necessary assumptions in §5.1 to get the $\mathsf{N}, L$, and the analytical regret upper bound. Then we demonstrate that RCB simultaneously satisfies the CBIC constraints in the whole decision process in §5.2 when sample size $N$ and probability $L$ are well designed. In §5.3, we show RCB achieves a $\mathcal{O}(\sqrt{KdT})$ regret.

### 5.1 REGULARITY CONDITIONS

In order to satisfy the CBIC constraint, we list two assumptions over the prior distribution.

**Assumption 1** (Prior-Posterior Distribution Assumption). *Denote* $G_t(i) = \min_{j \in B_t, i \in [K]/B_t} \mathbb{E}[\mu(x_t, i) - \mu(x_t, j)|S_{B_t}]$ *as the minimum prior-posterior gap when we have* $\mathsf{N}$ *samples of arm* $j \in B_t$ *and zero sample of arm* $i$ *in the cold start stage. There exists time-independent prior constants* $n_{\mathcal{P}}, \tau_{\mathcal{P}}, \rho_{\mathcal{P}} > 0$ *such that* $\forall n \geq n_{\mathcal{P}_0}, i \in [K]$, *then* $Pr(G_t(i) \geq \tau_{\mathcal{P}_0}) \geq \rho_{\mathcal{P}_0}$.

*Any given arm* $i$ *can be a posteriori best arm by margin* $\tau_{\mathcal{P}_0}$ *with probability at least* $\rho_{\mathcal{P}_0}$ *after seeing sufficiently many samples from* $B_t$. *The platform provides a fighting chance for those arms from* $[K]/B_t$ *with a low prioriori mean, which means after seeing sufficiently many samples of arm* $j \in B_t$ *there is a positive probability that arm* $i \in [K]/B_t$ *(zero sample collected) is better. What's more, we assume the gap between arms are at least greater than* $\tau_{\mathcal{P}_*}$ *with at least probability* $\rho_{\mathcal{P}_*}$ *after we have* $n_{\mathcal{P}_*}$ *data.*

**Assumption 2** (Posterior Distribution Assumption). *Denote* $G_t(b_t) = \min_{j \neq b_t} \mathbb{E}[\mu(x_t, b_t) - \mu(x_t, j)|S]$ *as the minimum posterior gap when we have* $\mathsf{N}$ *samples of each arms in the exploitation stage. There exist a uniform time-independent posterior constants* $n_{\mathcal{P}_*}, \tau_{\mathcal{P}_*}, \rho_{\mathcal{P}_*} > 0$ *such that* $\forall n \geq n_{\mathcal{P}_*}, i \in [K]$, *then* $Pr(G_t(b_t) \geq \tau_{\mathcal{P}_*}) \geq \rho_{\mathcal{P}_*}$.

The we provide the regularity conditions over covariates $\mathcal{P}_X$ as follows to avoid the singularity.

**Assumption 3** (Minimum Eigenvalue of $\Sigma$). *Define the minimum eigenvalue of the covariance matrix of $X$ as* $\lambda_{\min}(\Sigma) = \lambda_{\min}(\mathbb{E}_{x \sim \mathcal{P}_X}[xx^{\mathsf{T}}])$. *There exists such a* $\phi_0 > 0$ *satisfying that* $\lambda_{\min}(\Sigma) \geq \phi_0$.

**Assumption 4** (Prior Covariance Matrix Minimum Eigenvalue Assumption). *For each arm $i$, the minimum eigenvalue of prior covariance matrix $\Sigma_{i,0}$ satisfying: (1) $\Sigma_{i,0} \succeq \lambda_{i,0}\mathbf{I}_d$. (2) $\{\lambda_{i,t}\}_{t \geq 0}$ is increasing with order $\mathcal{O}(t)$.*

This assumption assumes that with more interaction and feedback occurred in the platform, users have a context-aware prior belief and this prior becomes weaker and weaker since users tend to trust the platform's recommendation rather than have strong belief for specific arms. And these minimum eigenvalues of the covariance matrix become larger which means that users are more open to those products rather than with strong opinion towards specific products. We also explore when this assumption is violated in Appendix §E.

### 5.2 CONTEXT-AWARE BAYESIAN INCENTIVE COMPATIBLE CONSTRAINT

Next we provide the requirements for the minimum sample size $\mathsf{N}(\epsilon)$ and the exploration probability $L$ to efficiently allocate the budget $\epsilon$ and effectively recommend the optimal arms to users.

**Theorem 1.** *With Assumptions 1 - 3, and the prior follows the normal distribution, if the parameters* $\mathsf{N}, L$ *are larger than some prior-dependent constant and the platform follows the RCB algorithm, then it preservers the $\epsilon$-CBIC property with probability at least $\rho_{\mathcal{P}_0}\rho_{\mathcal{P}_*}$. More precisely, it suffices to take*

$$\mathsf{N}(\epsilon) \geq \frac{(\sigma^2 d + 1)K^3}{\phi_0(\tau_{\mathcal{P}_*} + \epsilon)^2} \text{ and } L \geq 1 + \frac{1 - \epsilon}{\tau_{\mathcal{P}_0}\rho_{\mathcal{P}_0} + \epsilon}. \tag{6}$$

*And the exploitation stage starts at* $m_0(\epsilon) \geq \lceil 2 + \log_2 \mathsf{N}(\epsilon) \rceil$.

This theorem demonstrates that RCB maintains $\epsilon$-CBIC throughout the entire recommendation process given the lower bound of $\mathsf{N}$ and $L$. We provide that the minimum sample size $\mathsf{N}(\epsilon)$ is cubic with respect to the number of arms $K$, linear in relation to the covariate dimension $d$, inversely quadratic to the sum of budget $\epsilon$ and the minimal optimal posterior gap $\tau_{\mathcal{P}_*}$, and inversely linear to the minimum eigenvalue of the covariance matrix of our features $\phi_0$. This critically shows the tradeoff that a relatively larger budget $\epsilon$ significantly reduces the minimal sample size needed. Additionally, the

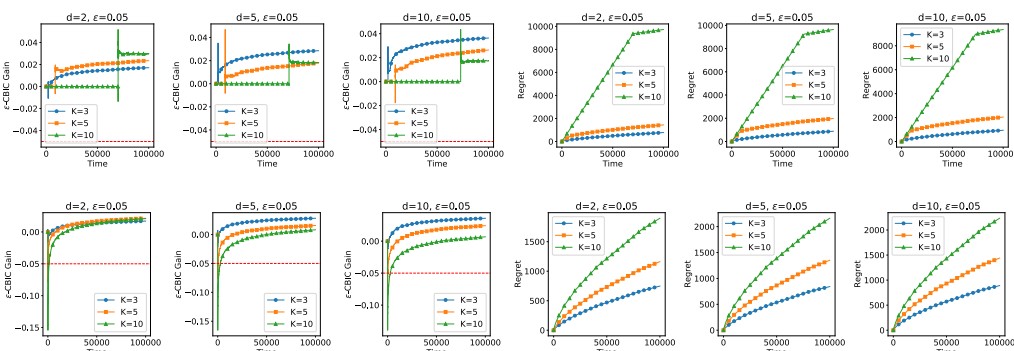

Figure 1: Incentive gain (left) and cumulative regret (right) of Setting 1 (upper) and Setting 2 (lower).

determination of the spread parameter $\gamma_m$ is based on the pivot of the functionality of $\epsilon$ in $\mathsf{N}(\epsilon)$. In RCB, given $\mathsf{N}(\epsilon)$ in the cold start stage, $\gamma_m$ for each epoch is entirely determined by the offline learning method and is independent of $\epsilon$ due to the increasing length of the epochs.

### 5.3 REGRET UPPER BOUND

In the following theorem, we show the regret upper bound of RCB.

**Theorem 2.** *Given* $\mathsf{N}(\epsilon)$ *and* $L$ *from Theorem 1, and Assumption 4, for any* $T \geq \tau_{m_0-1} + 1$, *with probability at least* $1 - \delta$, *the regret upper bound of RCB is* $\tau_{m_0-1}(\epsilon) + \mathcal{O}(\sqrt{Kd(T - \tau_{m_0-1}(\epsilon))})$.

The total regret is partitioned into two components: the cold start stage's regret $\tau_{m_0-1}$ and the exploitation stage $\mathcal{O}(\sqrt{KdT})$, where the latter depends only on the square root of the number of arms $K$, the covariate dimension $d$, and the decision horizon $T$. This square root dependency on $T$, $d$, and $K$ underscores the efficiency of the approach, as detailed in (Lattimore & Szepesvári, 2020). Moreover, the effect of the $\epsilon$ budget is predominantly observed in the regret of the cold start stage, especially when $T$ is small.

## 6 EXPERIMENTS

In this section, we apply RCB to synthetic data (§6.1) and real data (§6.2) to demonstrate its effectiveness by illustrating how RCB ensures sublinear regret, maintains CBIC, and exhibits robustness across various hyperparameters. Our code is available to ensure reproducibility of the results.

### 6.1 SIMULATION STUDIES

The goal of this section is to demonstrate that RCB algorithm can satisfy the $\epsilon$-CBIC constraint and simultaneously secure the sublinear regret. For all settings, the following parameters need to be specified (a) *environment parameters*: time horizon $T$, number of arms $K$, feature dimension $d$, and noise level $\sigma$; (b) *$\epsilon$-CBIC parameters*: budget $\epsilon$, prior-posterior minimum gap constants $\tau_{\mathcal{P}_0}$ and $\rho_{\mathcal{P}_0}$; (c) *prior belief parameters*: prior $\mathcal{P}_0$, where we assume the prior follows the normal distribution.

**Setting 1 (Environment Effects)**: We consider RCB's robustness in terms of different $K = [2, 5, 10]$, $d = [3, 5, 10]$. For rest parameters, we set $T = 10^5$, $\sigma = 0.05$, $\epsilon = 0.05$, $\tau_{\mathcal{P}_0} = 0.01$, and $\rho_{\mathcal{P}_0} = 0.95$. The prior are set to be $\beta_{i,0} = \mathbf{0}_d$ and $\Sigma_{i,0} = 1/5\mathbf{I}_d$.

**Setting 2 (Ad-hoc Design)**: This scenario demonstrates the results when the platform adopts an ad-hoc approach to $\mathsf{N}(\epsilon)$ without following the guidelines of Theorem 1. Here, $\mathsf{N}$ is set to $\{10, 100, 1000\}$. All other parameters remain consistent with those specified in Setting 1.

**Analysis of Setting 1 (Upper part of Figure 1)**: Different columns in the figure represent various dimensions $d$, with the first three columns illustrating the $\epsilon$-CBIC gain and the last three columns detailing the regrets observed. Our findings indicate that RCB satisfies the $\epsilon$-CBIC property, as evidenced by the gain consistently exceeding -0.05 (dashed line), or budget not been used up. During the exploitation stage, there is an observable upward trend in the instantaneous $\epsilon$-CBIC gain, suggesting that the recommendation system increasingly gains trust from customers (larger $\epsilon$ gain). The

right segment of the figure explores the relationship between regret, $d$, and $K$. It was observed that the regret for $K = 10$ significantly exceeds that for $K = 3$ and $K = 5$. This discrepancy arises because, to maintain the $\epsilon$-CBIC property, the duration of the cold start stage increases cubically with $K$, representing a substantial cost during this initial phase. In contrast, the impact of $d$ on cost is relatively minimal, as articulated in Theorem 1.

**Analysis of Setting 2 (Lower part of Figure 1):** This setting mirrors Setting 1 in terms of overall configuration. However, in this scenario, the platform does not adhere to the sample size requirements needed to satisfy the $\epsilon$-CBIC property, opting instead for an arbitrary fixed cold start length of $N(\epsilon) = \{10, 100, 1000\}$. The simulation results for $N(\epsilon) = \{100, 1000\}$ are detailed in Appendix §E. When compared with the regret observed in Setting 1, which is at the level of $10^5$, the regret in Setting 2 is considerably lower, at approximately $10^3$. However, in terms of $\epsilon$-CBIC gain, Setting 1 consistently shows positive gains, fully complying with the $\epsilon$-CBIC property, whereas Setting 2 experiences periods of negative gains, particularly when the number of arms is high ($K = 10$). This negative trend is more pronounced as $d$ increases, making it increasingly challenging to estimate an appropriate cold start length, as further discussed in Appendix §E. Notably, even with $N(\epsilon) = 1000$, the $\epsilon$-CBIC gain remains negative for most instances when $d = 5$ or $10$.

## 6.2   REAL DATA

We utilize a publicly available dataset from the Pharmacogenomics Knowledge Base (PharmGKB) that includes medical records of 5,700 patients treated with warfarin across various global research groups (Consortium, 2009). In the U.S., inappropriate warfarin dosing leads to about 43,000 emergency department visits annually. Traditional fixed-dose strategies can result in severe adverse effects due to initial dosing inaccuracies. Our study aims to optimize initial dosages by leveraging patient-specific factors from the cleaned data of 5,528 patients. Detailed data information and preproc are provided in Appendix E.2.

**Arms Construction:** We follow the arm construction as it in (Bastani & Bayati, 2020) and formulate the problem as a $K$-armed bandit with covariates ($K = 3$). We bucket the optimal dosages using the "clinically relevant" dosage differences: (1) Low: under 3mg/day (33% of cases), (2) Medium: 3-7mg/day (54% of cases), and (3) High: over 7mg/day (13% of cases). In particular, patients who require a low (high) dose would be at risk for excessive (inadequate) anti-coagulation under the physicians medium starting dose.

**Reward Construction**: For each patient, the reward is set to 1 if the dosing algorithm selects the arm corresponding to the patient's true optimal dose; otherwise, the reward is 0. This straightforward reward function allows the regret to directly quantify the number of incorrect dosing decisions. Additionally, it is important to note that while we employ a binary reward for simplicity, we model the reward as a linear function. Despite this, RCB demonstrates robust performance in this setting, indicating its applicability for scenarios involving discrete outcomes.

**Ground Truth:** We estimate the true arm parameters $\beta_i$ using the linear regression with the entire dataset for specific group. Besides, we scale the optimal warfarin dosing into $[0, 1]$ with minimum dosing as 0, and maximum dosing as 1. The true mean warfarin dosage is obtained from the inner production of $\beta_i$ (based on the optimal arm) multiples the covariate of this patient. Besides, for the counterfactual arm, the true mean dosage are set to be 0.

**RCB Setup**: The total number of trials is set at $T = 5528$, with reward noise $\hat{\sigma} = 0.054$ estimated from the true optimal dosing of warfarin after scaling. To create an online decision-making scenario, we simulate the process across 10 random permutations of patient arrivals, averaging the results over these permutations. The exploration budget $\epsilon$ is varied among $[0.025, 0.035, 0.045]$. The minimum gap $\tau_{\mathcal{P}_0}$ is set at 0.005. The prior variance is defined as $\Sigma = [0.4, 0.6, 0.8]\mathbf{I}_d$, and the prior means are $\beta_{2,0} = 0.05 \times \mathbf{I}_d$, $\beta_{1,0} = \beta_{3,0} = \mathbf{0}_d$. Further details on hyperparameters are available in §E.2.

**Evaluation Criteria:** We apply four criteria to evaluate the warfarin dose decision. (1) *Regret*: The regret is optimal mean dose minus 0. (2) $\epsilon$-*CBIC Gain*. (3) *Fraction of Incorrect Decision*: the fraction of incorrect decision. (4) *Weighted Risk Score*: the correct decision deserves 1 point and incorrect decision loss 1 point and multiple the true dosage sample proportion, which is newly proposed by us.

**Result Analysis:** In Table 1, we exhibits the RCB's true dosage correction ratio and physician assigned dosage correction ratio (always choose medium) and the weighted risk score.

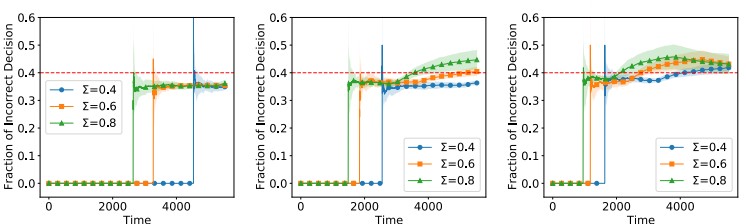

Table 1: Comparison `RCB` and physician algorithm and distribution of patients

| | | RCB Algo Assigned Dosage | | | Physician Algo Assigned Dosage | | | % of Patients |
|---|---|---|---|---|---|---|---|---|
| | | Low | Medium | High | Low | Medium | High | |
| **True Dosage** | Low | **50%** | 48% | 2% | **0%** | 100% | 0% | **27%** |
| | Medium | 14% | **84%** | 2% | 0% | **100%** | 0% | **60%** |
| | High | 2% | 93% | **5%** | 0% | 100% | **0%** | **13%** |

Figure 2: Left to right: fraction of incorrect decision under different setup of budgets ($\epsilon$) $[2.5, 3.5, 4.5] \times 10^{-2}$. Dotted line represents the lasso bandit's error rate (Bastani & Bayati, 2020).

**Fraction of Incorrect Decision**: In Figure 2, we present the fraction of incorrect decisions, a newly metric, which is particularly relevant in the field that the non-optimal arm has high cost and the optimal arm often remains unknown and difficult to ascertain. Our findings indicate varying levels of incorrect decisions based on the size of $\epsilon$ and different prior variances. At $\epsilon = 0.025$, three prior variances show a similar fraction of incorrect decisions, with all variations approximately at a 0.35 decision error rate, which is considered state of the art when compared to the lasso bandit described in (Bastani & Bayati, 2020), which utilizes prior knowledge of non-zero feature counts. At $\epsilon = 0.035$, only $\Sigma = 0.4\mathbf{I}$ achieves the lowest fraction of incorrect decisions, approximately 0.37. When $\epsilon$ is increased to 0.045, the fraction of incorrect decisions for all three beliefs exceeds 0.4. These observations suggest that with strong prior knowledge of the optimal dosage, a smaller $\epsilon$ improves correction rates. This highlights that `RCB` may require an extended cold start phase to reach optimal performance and build sufficient confidence in its recommendations.

**Weighted Risk Score**: In Table 1, we present the dosages assigned by `RCB`, the true dosages, the dosages assigned by a typical physician, and the true percentage of patients for each dosage. Notably, 60% of patients require a medium dosage, while 27% should receive a low dosage, and 13% a high dosage. We use blue percentages to indicate the correction rate of dosages assigned by `RCB` within each true dosage, and red percentages to denote extremely incorrect decisions across these levels. The physician algorithm, which consistently prescribes a medium level dosage, achieves a 100% correctness rate at the low dosage level. Conversely, `RCB` attains correction rates of 50%, 84%, and 5% for the low, medium, and high dosage levels, respectively, with an *extremely* incorrect rate of 2% for the low and high levels. With respect to the weighted risk score, we find that at $\epsilon = 0.025$, the three prior beliefs achieve scores of 0.291, 0.289, and 0.274, respectively, indicating higher scores are better. When $\epsilon = 0.035$ and $\Sigma = 0.4\mathbf{I}$, the score is 0.265. The physician policy, evaluated under the metric of the weighted risk score, calculates as $-1 \times 0.27 + 1 \times 0.60 - 1 \times 0.13 = 0.20$, significantly lower than the scores provided by `RCB` (0.291).

## 7    CONCLUSION

We propose a new `RCB` framework to address the context-aware BIC problem, where the information about the arms needs to be learned. This approach can leverage any sample-efficient machine learning method. We theoretically prove that `RCB` is regret-optimal in terms of the number of arms $K$, dimension $d$, and horizon length $T$, all in square root order, and satisfies the $\epsilon$-BIC constraints. Furthermore, we experimentally demonstrate that our algorithm achieves sublinear regret, is robust to different priors, dimensions, and budgets, and outperforms the state-of-the-art bandit algorithms.

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
