# OpenReview forum: "Context-Aware Online Recommendation with Bayesian Incentive Compatibility"
_ICLR.cc/2025/Conference — ICLR 2025 Conference Withdrawn Submission_

### Official Review · Reviewer_An4a · 2024-11-04

**Soundness:** 3
**Presentation:** 2
**Contribution:** 2
**Rating:** 3
**Confidence:** 2

**Summary:**

This paper proposed a bandit algorithm for context-aware online recommendation with Bayesian incentive compatibility. The algorithm is partitioned in two stages: exploration and exploitation. Regret bound is proven. Empirical results on simulated data and a real dataset of the warfarin dosage problem is presented to validate the effectiveness of the algorithm and theoretical results.

**Strengths:**

Proposed a novel bandit algorithm trading off exploration vs exploitation with regret bound in a simplified recommendation system setup.

**Weaknesses:**

The paper is motivated by online recommendation systems in internet economies, but I’m having a hard time trying to match the problem setup (as described in line 141 to 146) to any modern industrial online recommendation systems, such as ads, videos/movies or products etc. The application on the warfarin dosing problem is also confusing to me (see questions below). I wonder if this paper would be better appreciated in a more theory-focused venue such as COLT.

**Questions:**

For warfarin dosing problem, seems current physicians’ best practice is to start with a fixed dose (e.g., 5mg/day) then adjust based on patients’ reaction in a few weeks and converge to the optimal dose, and seems the key challenge is to determine the initial dosage, which, if off by too much, can cause stroke or internal bleeding. Bandit is an iterative algorithm, so I’m confused how this algorithm can help learn to determine the initial dose based on user interaction? Or the claim is that your proposed algorithm can converge faster than physicians?

Could you also elaborate a bit more on what would be “nonmyopic behavior” of users be? In a typical online recommendation setting, a user sends a request (e.g., by scrolling facebook page), and the platform, returns a list of “recommended” items (e.g., ads), the user either like the items (e.g., by clicking into it or even place an order) or not like it (e.g., by just ignoring it), so what does it mean for a user “not adhere to the platform's recommendations”? And the platform learns from the feedback and updates internal state for next recommendations, trading off exploration (recommend items highly uncertain if users would like to discover users’ new interests) vs exploitation (recommend similar items user liked before). So why would there be users’ myopic or nonmyopic behavior involved?

Seems the LASSO bandit by Bastani & Bayati (2020) is one main reference for the warfarin dosing problem; wonder if it makes sense to compare with the algorithm proposed in that paper?

line 091: what is “decision length T-free algorithm”?

---

### Official Review · Reviewer_JPHx · 2024-11-04

**Soundness:** 2
**Presentation:** 2
**Contribution:** 2
**Rating:** 3
**Confidence:** 3

**Summary:**

This paper formalizes CBICRP, which claims to formalize the exploration vs. exploitation tradeoff and Bayesian incentive compatibility into a Context-aware Bayesian Incentive-Compatible Recommendation Problem. A two-stage algorithm called RCB, which integrates incentivized exploration with an efficient offline learning component for exploitation. In the first stage of RCB, available products are explored while maintaining context-aware Bayesian incentive compatibility to determine sufficient sample sizes. The second stage of RCB employes inverse proportional gap sampling integrated with arbitrary efficient machine learning method to ensure sublinear regret. Experiments are conducted to validate the performance of the proposed algorithm.

**Strengths:**

This paper studies an important problem having theoretical value and potential application value.

The proposed algorithm is complemented with theoretical analysis and experiments.

**Weaknesses:**

1. The motivation is unclear. First, the motivation stated in the abstract is too broad for a technical paper. Exploration vs. exploitation tradeoff in recommendation applications has been studied extensively. Many approaches were proposed. Thus, using this tradeoff as motivation is vague and makes the readers confused. Second, I am not convinced that Bayesian incentive compatibility is a common problem for recommendation applications? To me incentive compatibility looks more natural, and the term “Bayesian” is more like a modeling approach. Also, the introduction does not provide any evidence and reference for it. Third, after reading the introduction, I do not get what technical problems this paper tries to address. Stating the high-level description of two obstacles is not enough to deliver the technical questions as well as the technical challenges.

2. This paper is not placed clear. The related work section simply enumerates previous works, but it does not clearly state how this work advance the literature.

3. This paper is hard to read. A lot statements and assumptions are stated without justification or evidence. I do not understand definition 1. Where is j come from in Equation (1)? What’s the formal formula of the optimal policy \pi*? The notation system is also poor. For example, in the definition the policy takes the sigma algebra as a parameter, while in Eq. (2), the sigma algebra disappears. The key challenge stated in the end of Section 3 is vague and broad.

**Questions:**

See weakness.

---

### Official Review · Reviewer_RB4a · 2024-11-04

**Soundness:** 2
**Presentation:** 2
**Contribution:** 3
**Rating:** 5
**Confidence:** 3

**Summary:**

This paper presents a recommendation problem, which aims to provide the optimal recommendation for users based on the users' context and explore more lesser-known products highly related to the system's context. The paper addresses two major challenges in this type of recommender system: (1) the exploration-exploitation trade-off in balancing new product exploration against exploiting known preferences, and (2) context-aware Bayesian incentive compatibility in accounting for users’ heterogeneous preferences and self-interested behaviors by proposing a two-stage algorithm. They proved their methods obtain sublinear regret and satisfy the Bayesian incentive compatibility ($\epsilon-$CBIC).

**Strengths:**

* The method is problem-driven and would have the potential for a wide application.

* The theoretical results are abundant and sufficient.

* The algorithms are well explained, guaranteeing reproductivity.

**Weaknesses:**

* The illustration of the theorems is somewhat unclear. Some notations are undefined before use (in Sections 3 and 4), such as $\tau_{\mathcal{P}_0}$ in Assumption 1. The notations and theorems lack practical explanation. I am confused how theorem 1 and 2 are straightforwardly connected to a better recommendation.

* The experiments are insufficient, where only a trivial baseline (Physician algo assigned dosage) is compared.

* Some connections between their asserted goals and their proposed methods are lacking, such as: the authors have mentioned the exploration-exploitation trade-off as a major challenge but I didn't see if their method can achieve this goal (obtaining pareto optimal points or not) in the present paper.

* In the cold-start stage, how the proposed method obtains an estimation for the posterior mean remains unknown. In this stage, the method only collects new data but does not update the parameters such as the $\beta_i$. How to obtain a varying posterior  based on this algorithm? Are there any parameters updated? It needs more details.

**Questions:**

*  The authors have said that their methods are robust against different priors, but this is not supported by their experimental results (only single priors are applied)

*  What is the practical meaning of $\tau_{\mathcal{P}_0}$, $\rho_{\mathcal{P}_0}$ and $\tau_{m_0-1}$ in Theorems 1 and 2? This section needs more careful checks to make readers clear about the details or the authors' main messages.

*  How did the proposed method address the exploration-exploitation trade-off?

*  The experiments are insufficient. The authors only compare their method with the lasso bandit (Bastani & Bayati, 2020) in their experiments, and Table 1 compares the method with the Physician Algo (which I think is a naive assignment method). It didn't support the conclusion that this method outperforms SOTA.

*  As shown in the fourth weakness, the cold-start stage is a bit unclear to me. How to obtain the posterior mean for $a_t^{\ast}$? Are the parameters updated in this step or only prior $\beta_i$ is applied thorough the whole stage? If so, it could be irrational as the prior might be extremely improper and affect the effectiveness of the cold-start stage.

* I cannot see the Appendix (as many contents are put into the Appendix as explained in the paper).

---

### Official Review · Reviewer_4ofR · 2024-11-04

**Soundness:** 3
**Presentation:** 2
**Contribution:** 2
**Rating:** 5
**Confidence:** 3

**Summary:**

This paper explores a recommendation problem aimed at providing optimal suggestions for users by considering their contextual features while also exploring novel products. It identifies two key challenges faced by such recommender systems: (1) The exploration-exploitation trade-off, which involves finding a balance between discovering new products and leveraging existing user preferences. (2) Context-aware Bayesian incentive compatibility, which addresses the diverse preferences and self-interested behaviors of users. To address these issues, the authors introduce a two-stage algorithm with the first stage collecting data as the cold-start stage, and the subsequent exploitation stage. The experiment results show that this approach meets the criteria for Bayesian incentive compatibility and outperforms the other methods with better recommendations.

**Strengths:**

S1: The Research questions are critical, such as the exploration-exploitation trade-off, and how to account for user heterogeneity

S2: The proposed algorithmic framework is reasonable, decomposing the training process into a cold-start period and an exploitation period, which is intuitional.

S3: The proposed theory shows the control of the upper bound for regret and for $\epsilon$-CBIC.

S4: The experiment results show that this approach meets the criteria for Bayesian incentive compatibility and outperforms the other methods.

**Weaknesses:**

W1: My major concern is how this approach can achieve exploration and exploitation trade-off. In the cold-start phase, the Promoted Recommendation part and the Organic Recommendation part are not novel and are not enough for the trade-off

W2: There are many unclear presentations, such as many symbols is missing definitions ($I_t$ in line 141, OffPos in algorithm 2), some misuse notations ($L$ can be both the upper bound of ||x|| and the probability), and the Appendix is missing.

W3: Lack of baselines. I think there exists more baselines for the exploration and exploitation trade-off, the authors should add some in the experiments.

**Questions:**

Q1: If all the arms are pulled N times, will it end the cold start phase? In addition, should the "N samples" in line 258 be "at least N samples"?

Q2: In line 253, how can we compute the posterior? Because there seems no parameter update in the cold-start phase. In addition, should we consider the mixture distribution of prior and posterior instead of the hard truncation in line 253?

Q3: Is $j$ arbitrary in Definition 1?

---

### Note · Authors · 2024-11-26

I have read and agree with the venue's withdrawal policy on behalf of myself and my co-authors.